# Senescent Cells: Dual Implications on the Retinal Vascular System

**DOI:** 10.3390/cells12192341

**Published:** 2023-09-23

**Authors:** Mohammad Reza Habibi-Kavashkohie, Tatiana Scorza, Malika Oubaha

**Affiliations:** 1Department of Biological Sciences, Université du Québec à Montréal (UQAM), Montréal, QC H2L 2C4, Canada; mr.habibi7388@gmail.com (M.R.H.-K.); scorza.tatiana@uqam.ca (T.S.); 2The Center of Excellence in Research on Orphan Diseases, Courtois Foundation (CERMO-FC), Montreal, QC H3G 1E8, Canada

**Keywords:** cellular senescence, senescence-associated secretory phenotype (SASP), retinal vascular system, proliferative retinopathies, senotherapeutic agents

## Abstract

Cellular senescence, a state of permanent cell cycle arrest in response to endogenous and exogenous stimuli, triggers a series of gradual alterations in structure, metabolism, and function, as well as inflammatory gene expression that nurtures a low-grade proinflammatory milieu in human tissue. A growing body of evidence indicates an accumulation of senescent neurons and blood vessels in response to stress and aging in the retina. Prolonged accumulation of senescent cells and long-term activation of stress signaling responses may lead to multiple chronic diseases, tissue dysfunction, and age-related pathologies by exposing neighboring cells to the heightened pathological senescence-associated secretory phenotype (SASP). However, the ultimate impacts of cellular senescence on the retinal vasculopathies and retinal vascular development remain ill-defined. In this review, we first summarize the molecular players and fundamental mechanisms driving cellular senescence, as well as the beneficial implications of senescent cells in driving vital physiological processes such as embryogenesis, wound healing, and tissue regeneration. Then, the dual implications of senescent cells on the growth, hemostasis, and remodeling of retinal blood vessels are described to document how senescent cells contribute to both retinal vascular development and the severity of proliferative retinopathies. Finally, we discuss the two main senotherapeutic strategies—senolytics and senomorphics—that are being considered to safely interfere with the detrimental effects of cellular senescence.

## 1. Cellular Senescence

### 1.1. Cell Senescence and Inducers

Throughout life, cells are constantly challenged by internal and external stressors. Repair, cell death, and senescence are the main physiological programs adopted by tissue cells to respond to triggers, depending on the severity and nature of stress. Cellular senescence is a state of irreversible cell cycle arrest, with a preserved metabolic function, that is stimulated in normal cells in response to various internal or external stress/stimuli, as well as developmental signals [1]. The first evidence of senescence was introduced in 1961 when Leonard Hayflick and Paul Moorhead discovered limited proliferation capacity in human fibroblasts after extensive serial passaging [2]. According to the nature of stimuli, cellular senescence is classified into two categories: replicative and stress-induced senescence. Replicative senescence (RS) is triggered by telomere attrition, whereas stress-induced premature senescence (SIPS) is caused by stressors such as unresolved DNA damage, oxidative stress, activated oncogenes, irradiation, mitochondrial dysfunction, genotoxic drugs, cell–cell fusion, epigenetic modifiers, and impaired proteostasis [3]. An intriguing aspect of cellular senescence is that although characteristically occurring in injured tissues, in which it is transient and important for tissue regeneration, it increases with aging and happens during embryonic development. Thus, cellular senescence is found in homeostatic biological processes of very distinct etiology.

### 1.2. Cell Cycle Arrest

An active cell cycle in multicellular eukaryotes is divided into four distinct phases: G1 (a decision window to enter or exit the cell cycle), S (DNA duplication), G2 (a decision window to initiate or cease the process that leads to chromosome segregation), and M (DNA segregation) [4]. Cell cycle progression is driven by phase-specific cyclin-CDK complexes that phosphorylate important regulatory targets. In the pre-replicative G1 phase, the accumulation of cyclin D-CDK4/6 complexes commits cells to enter the next cell cycle, thereby inhibiting cell cycle exit (G0 state) [4]. Indeed, there is a decision window here that allows cells to continue or exit the cell cycle transiently or permanently. Subsequent accumulation of cyclin E-CDK2 inhibits retinoblastoma protein (RB), leading to cyclin A-CDK2 accumulation, replication initiation, and S phase entry [4]. After S phase completion, the entry into mitosis and the ${APC/C}^{Cdc20}$-mediated degradation of cyclins, which is necessary to complete the cell cycle, are driven by the activity of the cyclin A/B-CDK1 complex [4].

Cell cycle arrest, regardless of the cell type, is the primary feature of senescent cells. However, growth cessation is not an exclusive characteristic of senescent cells, being also observed in cellular quiescence and terminally differentiated cells.

***Terminally differentiated cells:*** The development, maintenance, and regeneration of many human tissues highly depend on the terminal differentiation of resident progenitor cells [5]. During terminal differentiation, undifferentiated progenitor cells undergo a gradual process of specialization and permanently exit the cell cycle upon reaching a fully differentiated post-mitotic state [6]. These highly specialized cells are refractory to pro-mitogenic signals and work together to facilitate the specialized functions of a tissue. Previous studies indicate that cells can undergo terminal differentiation and exit the cell cycle at G1 simultaneously, and this transformation is made possible by the intricate cooperation of multiple molecules, including CIP/KIP inhibitors, pRB, and ${APC/C}^{Cdh1}$ [5,6]. In contrast to other non-proliferating cells, such as those in quiescence and senescence, the morphology, metabolic rate, and transcriptome profile of terminally differentiated cells are significantly shaped by their typical cellular functions within a specific tissue. Neurons, muscles, adipocytes, and bone cells are some examples of terminally differentiated cells that exit the cell cycle permanently while adopting cell type-specific transcriptional programs [6].

***Cellular quiescence:*** Cell cycle arrest in quiescent cells is reversible, as it enables cells to return to the normal cell cycle in response to certain pro-mitogenic factors and signals. For instance, tissue injury stimulates local resident quiescent adult stem cells (qASCs) to re-enter the cell cycle, causing the production of transient progenitor and mature functional cells that promote repair and tissue regeneration [7]. In general, proliferation arrest in quiescent cells is mediated by cyclin-dependent kinase inhibitors (CDKIs) including ${p21}^{CIP1/WAF1}$ ${p27}^{KIP1}$, and ${p57}^{KIP2}$ at G0 [8,9]. Nonetheless, quiescent cells are not only restricted in terms of replication ability, but are also characterized by a reduced size, dense heterochromatin, low metabolic activity, and reversible suppression of global RNA and protein synthesis [10]. Quiescent cells satisfy their lower energy demand by producing ATP via glycolytic pathways and fatty acid oxidation, and therefore have lower oxidative phosphorylation and mitochondrial activity [10].

### 1.3. The Senescent Phenotype

Cell senescence is associated with dramatic changes in cell morphology, structure, and metabolism (Figure 1). The flattened, enlarged, and multinucleated morphology of senescent cells (SCs) reflects changes occurring in the number and function of membranous organelles such as lysosomes, mitochondria, endoplasmic reticulum (ER), nucleus, etc. Senescent cells, both in vivo and in vitro, are often characterized by multiple nuclei within an enlarged cytoplasm [11,12]. It has been proposed that multinucleated senescent cells can be generated through endomitosis/cytokinesis failure [13,14,15], cell–cell fusion [13], and a process referred to as “amitosis” (involving the fragmentation of polyploid nuclei during interphase) [16]. However, a previously published study by Dikovskaya et al. proposed a mechanism indicating that multinucleation in senescent cells occurs predominantly due to mitotic failure [17].

Increased lysosomal biogenesis and elevated autophagic activity are documented during the establishment of SCs [18]. Furthermore, the protein compositions of lysosomes are shown to change significantly upon senescence [18]. Lysosomal hydrolytic enzymes are required for the degradation of damaged organelles and proteins via the autophagic pathway. Lysosomal β-galactosidase is the most widely used biomarker for SCs and becomes detectable at a pH of 6.0 due to the increased lysosomal mass in affected cells [19]. However, the drawback of using this marker as a reporter of senescence is that SA-β-gal activity is also detected in non-senescent cells cultured under confluent or serum-starved conditions [20]. Lipofuscin is a complex fluorescent mixture of highly oxidized proteins and lipids that due to their cross-linked nature are non-degradable and accumulate in lysosomes, only being segregated during cell division [21,22]. Lipofuscin accumulation, which can be detected by Sudan Black B staining, is another hallmark of SCs.

Although the cell cycle is arrested in SCs, these cells are still metabolically active. At a senescent state, cells have higher metabolic demands due to their enlarged size and to their robust production of secreted proteins (SASP) [23]. Generally, cells produce units of energy (ATP) through mitochondrial oxidative phosphorylation (OXPHOS) and anaerobic glycolysis in the presence and absence of oxygen, respectively [24]. It has become apparent that during cellular senescence, the metabolism shifts from oxidative phosphorylation (OXPHOS) towards glycolysis even in presence of high oxygen levels [25,26,27]. Mitochondrial mass increases in SCs may partially be a consequence of the increased size of cells in senescence [28]. However, decreased mitophagy activity in senescence promotes the accumulation of dysfunctional mitochondria in SCs [28]. Although mitochondrial mass increases dramatically in SCs, they are characterized by a reduced respiratory capacity [28]. Reactive oxygen species (ROS) are a by-product of mitochondrial respiration and increase strikingly in SCs due to low mitochondrial membrane potential [28,29]. Mitochondria-derived ROS cause mitochondrial damage and DNA breaks at the telomeric regions leading to cell cycle arrest [29,30]. It has also been shown that cytosolic p53 accumulation in SCs concurrent to ROS generation in SCs results in Parkin inhibition (an important regulator of mitophagy) and mitochondrial dysfunction [31]. Generally, NADH molecules are oxidized in the mitochondria to NAD+. In mitochondrial dysfunction-associated senescence (MiDAS), accumulation of cytosolic NADH results in the inhibition of glycolytic enzymes, ATP depletion, and finally, cell cycle arrest [32].

Upregulation of SASP genes is one of the most important phenotypic changes occurring in SCs (Figure 1). The synthesis of SASP factors, which include a group of proinflammatory cytokines, chemokines, proteases, and growth factors, results in ER expansion in SCs. Excessive protein synthesis in the ER leads to the accumulation of misfolded proteins and activates the unfolded protein response (UPR) [33]. The ATF6α pathway of UPR is particularly responsible for increasing the size of the ER in SCs [34,35]. The chromatin structure of SCs also changes globally by the formation of senescence-associated heterochromatin foci (SAHF); the latter are specialized regions of facultative heterochromatin that suppress transcription of proliferation-promoting genes [36]. SAHF structures can be detected by DAPI and by antibodies specific to components of SAHF such as H3K9Me2/3, macroH2A, and HP1 proteins. However, SAHF are not a universal biomarker and appear to be induced in SCs by activated oncogenes and DNA replication stressors [37].

### 1.4. Senescence Molecular Pathways

During cellular senescence, growth arrest occurs through the activation of both the p53/p21(CIP1/WAF1) and p16INK4a/RB signaling pathways (Figure 2). In the p53-dependent senescence pathway, telomere attrition or elevated ROS levels, generated by intrinsic and extrinsic stimuli such as genotoxic stress and mitochondrial dysfunction, result in DNA damage and activation of the DNA damage response (DDR) pathway [38]. The DDR network involves different regulatory proteins including ATM/ATR, CHK1/CHK2, p53, and p21^(CIP/WAF1)^. ATR/CHK1 and ATM/CHK2 signaling are usually activated by single-strand and double-strand DNA breaks (DSBs), respectively [39,40,41]. p53 phosphorylation is induced in both pathways, which allows its binding to the CDKN1A promoter and upregulation of p21^(CIP/WAF1)^ expression. The latter induces senescence by inhibiting the kinase activity of cyclin-CDK complexes and RB phosphorylation [42]. In the p53-independent senescence pathway, the CDK4/6-CyclinD complexes are directly inhibited by p16^INK4a^ and lose their regulatory effects on RB phosphorylation (Figure 2). Dephosphorylated RB binds to the E2F and induces cellular senescence [43].

### 1.5. Autophagy: A Pro-Senescence or an Anti-Senescence Mechanism?

Cellular autophagy is known to decrease with aging, while SCs accumulate in tissues with advancing age [44]. Autophagy was initially considered a mechanism preventing cell senescence, through the elimination of damaged components (e.g., proteins or mitochondria) [45]. This assumption is supported by several studies demonstrating the anti-senescence effects of autophagy. For example, the re-establishment of basal autophagy activity in aged satellite cells prevents cellular senescence and restores the regenerative capacity of geriatric satellite cells [46,47]. Moreover, Kang et al. found that autophagy impairment in removing defective mitochondria induces cellular senescence in primary human fibroblasts in an ROS- and p53-dependent manner [48]. However, this overriding view has been challenged by several recently studies. Targeting essential genes for autophagy, ATG5 and ATG7 delayed the senescence state and reduced the production of the key SASP factors including IL-6 and IL-8 [49]. Further research has also underlined the contribution of a specialized type of autophagy called the TOR-autophagy spatial coupling compartment (TASCC), in the massive synthesis of the SASP factors during the establishment of oncogene-induced senescence (OIS) [50]. Additional evidence for the pro-senescence effects of autophagy has been provided by Nam et al. whose results demonstrated that prolonged activation of autophagy via mTOR inhibitors induces cellular senescence in radiation-resistant cancer cells [51]. These discrepancies may be explained by the type of autophagy involved (basal or induced), as well as by when and where autophagy acts [44,45]. However, the complex interplay between autophagy and cell senescence still needs further investigation to clarify the context-dependent implications of autophagy in cellular senescence.

## 2. Regulatory Mechanisms of the SASP

Complex signaling pathways are responsible for the downstream robust secretion of hundreds of bioactive molecules including proinflammatory cytokines, chemokines, growth modulators, angiogenic factors, proteases, bioactive lipids, extracellular matrix components, and matrix metalloproteinases (MMPs). Several studies suggest the involvement of SASP factors such as IGFBP3, IGFBP4, and IGFBP7 in driving paracrine senescence in nearby cells [52,53,54]. The SASP profile of SCs varies, depending on the cell type and the type of inducer [55,56]. Like pathways involved in senescence growth arrest, further studies are needed to elucidate mechanisms that regulate the SASP program. Here, we will review different pathways involved in the development and regulation of SASP.

### 2.1. NF-κB Signaling

NF-κB is a master regulator of SASP expression, being maintained inactive in the cytoplasm due to its binding to the inhibitory molecules of the IKB family [57]. The IKK kinase complex consists of two catalytic subunits (IKKα and IKKβ) and one regulatory subunit (NEMO/IKKγ). Genotoxic stress results in NEMO post-translational modification by ATM (ataxia–telangiectasia mutated), which in turn activates the IKK complex, leading to NF-κB translocation to the nucleus [58,59]. The NF-κB (p65p50 dimer) binds to DNA and stimulates the expression of SASP genes. Reduced expression of SASP genes following depletion of DDR components such as ATM, NBS1, or CHK2 supports the importance of DDR activation for robust SASP production [60]. In addition to DDR signaling, the IKK complex can be activated by various stimuli including oxidative stress, growth factor, pathogens, and proinflammatory cytokines [61].

SASP production is not an inevitable consequence of growth cycle arrest. A recent study by Coppé et al. reported a lack of paracrine activity in p16-induced SCs [57]. Indeed, ectopic expression of p16^INK4a^ or p21^(CIP/WAF1)^ arrested the cell cycle without inducing the SASP program [57]. In addition, a previous study by Freund et al. identified the sufficiency of p38MAPK signaling in driving the SASP program, independent of DDR signaling [58]. p38MAPK regulates the expression of SASP factors by inducing NF-κB activity [58].

### 2.2. cGAS/STING Signaling

Generally, DNA damage by external or internal stimuli may cause pathological accumulation of DNA fragments of nuclear origin in the cytoplasm [62,63]. In normal and healthy cells, cytoplasmic DNA fragments are eliminated by cytoplasmic DNases such as DNAase 2 and TREX1 (DNAase 3) [63,64]. In SCs, the expression of the DNAases is downregulated, leading to the accumulation of free DNA in the cytoplasm, and stimulation of the cGAS/STING signaling pathway [62]. Upon sensing aberrant double-stranded DNA molecules, the cytosolic DNA sensor cGAS activates the stimulator of the interferon genes (STING) via cGMP [61]. STING then recruits the complex of IKK kinase and promotes NF-κB translocation to the nucleus [1,65,66,67,68,69,70,71,72].

Recent studies have shown that the integrity of the nuclear envelope plays a critical role in regulating gene expression [73,74]. In SCs, the morphological alterations in the nuclear envelope may stimulate SASP expression in a cGAS/STING-dependent pathway [75,76,77]. The nuclear lamina in the inner nuclear membrane (INM) contributes to the size, shape, and stability of the nucleus. Declined expression of lamin B1 in SCs is concurrent with the activation of either the p53 or pRB pathways [73]. Indeed, lamin B1 degradation results in the formation of cytoplasmic chromatin fragments in SCs and the expression of SASP genes via the cGAS/STING signaling pathway [75,76,77]. STING is also activated by DNA damage in a cGAS-independent manner [78]. In non-canonical STING activation, the tumor suppressor p53 and the E3 ubiquitin ligase TRAF6 are involved in STING activation and preferentially activate NF-κB rather than IRF3 [78].

### 2.3. mTOR Signaling

The involvement of the mTOR-NF-κB axis in modulating the production of proinflammatory SASP factors by SCs has been recently investigated. A study conducted by Laberge et al. reported the selective suppression of SASP factors by mTOR inhibition via rapamycin, without reversing cell cycle arrest [79,80]. The translation of a subset of important mRNAs, such as IL1α, the master regulator of SASP subsequently engaging IL-6/IL-8, is highly dependent on the modulatory activity of mTOR signaling [81]. Autocrine or paracrine signaling of the translated factor, IL1α, via the IL1AR receptor activates NF-κB, which in turn stimulates the transcription of several SASP genes [79,81]. Harranz et al. have also shown the importance of MK2 (also known as MAPKAPK2), another mTOR-dependent transcript, in modulating the transcription of the SASP genes [80]. In fact, phosphorylation and inhibition of the RNA-binding protein ZFP36L via MK2 stabilize the mRNAs of many SASP transcripts [80,81]. Altogether, these findings suggest SASP inhibition by the mTOR inhibitor rapamycin, concurrent with suppressed IL1-α and MK2 translation [79,80]. Further research by Van Vliet et al. also demonstrated that a physiological hypoxic condition limits generation of SASP through suppression of the mTOR-NF-κB signaling loop [82].

### 2.4. C/EBPβ Signaling

The CCAAT/enhancer-binding protein (C/EBP) family consists of several regulatory factors that are involved in both the induction and suppression of SASP genes. Structurally, these factors are composed of a transactivation domain (TAD), a basic DNA-binding domain (DBD), a regulatory domain, and a “leucine zipper” domain [83]. According to previously published data, among the C/EBP factors, C/EBPβ and C/EBPγ may play opposite roles in regulating the expression of SASP genes. After dimerization, activated C/EBPβ factor binds to its specific DNA sequence via the DNA-binding domain (DBD) and stimulates the expression of several genes including IL-1β, IL-8, IL-6, the growth-regulated oncogene α (GROα or CXCL1), and neutrophil-activating protein 2 (NAP2, also CXCL7) [83,84]. In addition, C/EBPβ homodimers also play a pivotal role in driving Ras-induced premature senescence [85]. In contrast, the C/EBPβ:C/EBPγ heterodimer inhibits cellular senescence and suppresses the transcription of SASP genes [86]. Exceptionally, the structure of the C/EBPγ factor lacks the transactivation domain and is unstable in its homodimeric form [86]. C/EBPγ depletion is associated with declined proliferation and stimulation of senescence in tumor cells [86]. However, C/EBPβ depletion restores the growth of C/EBPγ-deficient cells, which confirms the importance of C/EBPγ as a growth-promoting transcription factor and negative regulator of cellular senescence [86].

### 2.5. NOTCH1 Signaling

The evolutionarily conserved NOTCH1 signaling pathway regulates SASP components via the inhibition of C/EBPβ [87]. At the early phase of RIS, the upregulation of NOTCH1 is necessary and sufficient for limiting C/EBPβ activity and for increasing the expression of both TGF-β1 and TGF-β3 in SCs [87]. Downregulation of the active form of NOTCH1 at the late phase of RIS rather elevates the expression of the inflammatory cytokines IL-1β, IL-6, and IL-8 via C/EBPβ transcription factor [87]. It has been also demonstrated that IL-1α is an upstream SASP effector which activates the C/EBPβ transcription factor to induce its downstream targets genes such as IL-1β, IL-6, and IL-8 [88,89]. Altogether, NOTCH1 signaling modulates IL-1α and expression of downstream SASPs via downregulation of the transcription factor C/EBPβ [87,88].

### 2.6. JAK-STAT Signaling

The JAK/STAT (Janus kinase/signal transducer and activator of transcription) pathway plays critical roles in a variety of biological events, such as immune fitness, hematopoiesis, tissue repair, apoptosis, adipogenesis, and inflammation [90]. The JAK family is composed of four members (including JAK1, JAK2, JAK3, and tyrosine kinase 2 (TYK2)) that act through downstream STAT proteins (including Stats 1–4, Stat5a, Stat5b, and Stat6) [91]. Compared to non-senescent cells, the JAK-STAT signaling pathway is highly active in SCs [91]. Inhibition of the JAK-STAT signaling pathway in SCs suppresses SASP expression and alleviates age-related tissue dysfunction [91]. Accumulation of senescent preadipocytes, fat cell progenitors, in aged adipose tissue develops a proinflammatory SASP and inflammation in the whole adipose milieu [92]. To investigate the role of the JAK-STAT pathway in age-related inflammation, Xu et al. performed a series of experiments and demonstrated that inhibition of the JAK-STAT signaling pathway suppresses the SASP in aged preadipocytes and human umbilical vein endothelial cells (HUVECs) [92]. Similarly, research performed by Chen et al. reported accumulation of senescent tendon stem/progenitor cells (TSPCs) in tendon tissue with advancing age [93]. The results from this study also demonstrated that increased JAK-STAT signaling induces senescence in tendon stem/progenitor cells [93]. Treatment of aged TSPCs with a JAK-STAT signaling pathway inhibitor (AG490) attenuated the senescence phenotype and restored age-related dysfunctions, including self-renewal, migration, actin dynamics, and stemness [93]. Multiple studies also demonstrated the contribution of the JAK-STAT pathway in developing severe symptoms such as a cytokine release syndrome in COVID-19 patients [94,95].

## 3. Physiological Roles of Cellular Senescence

Cellular senescence plays a pivotal tumor-suppressing role through a variety of physiological mechanisms, including cell cycle exit reinforcement, paracrine senescence induction, and immune cell recruitment [96]. At the physiological level, the transient secretion of SASP factors by SCs participates in the progression of several processes, some of which are highlighted below (Figure 3).

### 3.1. Embryogenesis

Cellular senescence contributes to the embryonic development of many species in a developmentally programmed manner. Embryogenesis is a complex multistep process involving cell migration, proliferation, differentiation, and apoptosis, that is tightly regulated in time and location. An exhaustive review of the literature linking programmed cellular senescence with homeostatic embryogenesis supports the possibility of an evolutionary origin of cellular senescence as a developmental mechanism [97,98,99,100]. The developmental role of cellular senescence was first identified in bird embryonic development, when Nacher et al. described SAβG activity in degenerating quail mesonephros [101]. Thereafter, investigation of SAβ-gal activity in chick embryos revealed its occurrence at multiple locations including the limbs, pharyngeal arches, neural tube, and developing eye [99]. During mouse embryonic development, SCs appear at embryonic day 9.5 and undergo apoptosis or are cleared by macrophages at embryonic day 17.5 [102,103]. Intense signals have been reported in various developing structures within mouse embryos, including the mesonephros, endolymphatic sac of the inner ear, apical ectodermal ridge (AER), as well as the neural roof plate, through whole-mount SAβG staining [98,99]. In adult tissues, SCs express known senescence markers including p53, ${p16}^{INK4a}$, p21^(CIP/WAF1)^, ${p19}^{ARF}$, β-galactosidase, and DNA damage [102,103]. Interestingly, in the developing fetus, SCs are negative for p53, ${p16}^{INK4a}$, ${p19}^{ARF}$, and for markers of DNA damage, but are positive for certain senescence markers, including p21^(CIP/WAF1)^, β-galactosidase, and SASP factors [98]. A recently published study by Munoz-Espin et al. shows that the developmentally programmed senescence in the mouse embryo is highly dependent on p21^(CIP/WAF1)^, but independent of p53 [98]. Indeed, p21^(CIP/WAF1)^ upregulation is mediated by the TGF-β/SMAD and PI3K/FOXO pathways in a p53-independent manner [98,99]. Furthermore, ${p16}^{INK4a}$-dependent senescence has been reported in a subpopulation of motoneurons in the developing spinal cord, as well as during placental development [104]. In summary, developmental SCs are characterized by a high expression of p21^(CIP/WAF1)^ and ${p16}^{INK4a}$ (cell cycle inhibitors) and contribute to the embryonic development of many species.

### 3.2. Regeneration

Some vertebrates like salamanders and zebrafish display remarkable regeneration capacity by fully restoring a lost limb after a severe injury [105]. Interestingly, the regeneration capacity of salamanders is not affected by repeated challenges or aging [106]. Research conducted by Yun et al. reported significant induction of cellular senescence during salamander limb regeneration, indicating its important role in driving tissue regeneration [107]. A lack of major changes in the number of SCs following multiple rounds of limb regeneration suggests a rapid and efficient mechanism for the clearance of SCs [107]. Selective elimination of macrophages via clodrosomes (clodronate salt-containing liposomes) results in the persistence of SCs and impaired limb regeneration in the salamander [107]. In fact, the transient presence of SCs and SASP factors stimulate the recruitment of immune cells, in particular macrophages, to amputated limbs, which leads to the elimination of SCs by these scavengers, following each round of limb regeneration [107,108]. However, many questions have remained yet regarding the mechanisms by which macrophages identify SCs and eliminate them but not normal cells [108].

As for the salamander, the regenerative capacity of zebrafish is not affected with age or following multiple limb amputation [109,110]. Transient induction of cellular senescence after pectoral fin amputation drives the regeneration process in zebrafish, considering that the removal of SCs by senolytic components including ABT-263 (Navitoclax) or quercetin impairs fin regeneration [111]. A recent study performed by Bednarek et al. also documented a transient induction of cellular senescence during zebrafish heart injury and regeneration [112]. Comparable regeneration of the heart and caudal fin in young and old fish supports the notion that the zebrafish maintains regenerative ability throughout its lifespan [110]. Upon amputation, the activated Wnt/β-catenin pathway regulates many aspects of fin regeneration [109]. By using a heat-shock inducible transgenic hsp70l:Dkk1-GFP fish model, Azevedo et al. reported disturbed regeneration and impaired blastema formation after Wnt/β-catenin inhibition [109]. However, the impaired regenerative capacity of the caudal fin was recovered by a new amputation at a non-inhibitory temperature. This indicates that blastema formation does not depend on stem/progenitor cells that are highly reliant on the Wnt/β-catenin signaling for their survival [109].

### 3.3. Wound Healing

Positive implications of SCs in wound healing have been demonstrated in multiple mammalian tissues such as the skin [113], liver [114], and heart [115]. Excessive fibrosis impairs normal tissue repair by decreasing tissue elasticity. The high expression of the matricellular protein CCN1 during cutaneous wound healing induces fibroblast senescence through an integrin-mediated mechanism, which curbs fibrosis during tissue repair [113]. The generation of senescent fibroblasts in granulation tissues requires p53 and ${p16}^{INK4A}$ mediators, as well as activation of the ERK and p38 MAPK pathways [113]. Further studies by Demaria and colleagues have revealed optimized myofibroblast differentiation and promoted cutaneous wound closure through early secretion of the SASP factor, PDGF-AA, by fibroblasts and endothelial cells [116].

In conditions of chronic liver damage, the excessive production of extracellular matrix components by activated hepatic stellate cells (HSCs) promotes liver fibrosis, which may lead to liver cirrhosis, a major public health problem worldwide [117]. Krizhanovsky et al. studied the biological implications of cellular senescence on liver fibrosis by using a p53;INK4a/ARF null mouse model and treating it with CCL4, a chemical agent known to induce liver fibrosis and cirrhosis [114]. In comparison with wild-type mice, liver samples derived from mice with compound mutation contained significantly lower senescent-activated HSCs and developed more severe liver fibrosis [114]. In fact, the limited proliferative and fibrogenic capacity of senescent-activated HSCs reduced fibrotic progression and liver cirrhosis. Furthermore, factors secreted by senescent HSCs recruit NK cells into fibrotic lesions, that preferentially eliminate SCs and contribute to fibrotic resolution [114]. In summary, in the liver, cellular senescence promotes wound healing by limiting the fibrogenic activity of HSCs and attracting immune cells to the injured region.

Meyer et al. reported an aggravated heart fibrosis and functional impairment after the genetic depletion of Trp53/Cdkn2α, suggesting senescent fibroblasts as a critical regulator of cardiac fibrogenesis [118]. Following myocardial infraction, hypoxia stimulates the senescence program in cardiac fibroblasts through a CCn1-dependent manner that eventually curbs collagen production and cardiac fibrosis [115,119]. In conclusion, it seems likely that the timely induction of SCs has cardioprotective effects upon a heart injury.

## 4. Retinal Vascular System

### 4.1. Retina

The retina is one of the most metabolically active neural tissues, with multiple layers of cells responsible for phototransduction. The highly specialized tissues of the retina are supplied by two vascular systems: (1) the inner retina supplied by the central retinal artery and (2) the retinal pigmented epithelium and outer retina supplied by the choriocapillaris. The intricate neuronal layers of the retina are composed of six different cell types including photoreceptors (rods and cones), bipolar cells, horizontal cells, amacrine cells, and retinal ganglion cells (RGCs) [120]. Studying cellular and molecular mechanisms involved in retinal vasculature development is pivotal for understanding the retinal vasculopathies and associated alterations in eye diseases.

### 4.2. Retinal Blood Vessel Development

In humans, retinal vascular development begins at the fourth week of gestation, during which hyaloid vessels appear and supply oxygen and nutrients to the developing inner retina [121]. The hyaloid vasculature regresses during mid-gestation, while the retinal vasculature develops simultaneously [121]. Retinal vascularization in humans starts at around 16 weeks of gestation and is typically completed before birth, usually around the 40th week of gestation [122]. Normal function of the retinal vascular system is necessary for the delivery of oxygen, nutrients, and hormones and the removal of carbon dioxide and waste products. Over the past several decades, studies on retinal vasculature development have greatly expanded our knowledge about the cellular and molecular alterations involved in the pathogenesis of ocular diseases. Generally, vascular development happens by two functionally distinct processes referred to as vasculogenesis and angiogenesis.

### 4.3. Vasculogenesis

Vasculogenesis refers to the differentiation of angioblasts into endothelial cells (ECs) and de novo assembly of a primitive vascular network [123]. Angioblasts are derived from the mesoderm and stimulated by fibroblast growth factors (FGFs) [124]. Recent evidence shows that vasculogenesis is not limited just to the period of embryonic development, but is also involved in blood vascularization in ischemic, malignant, and inflamed tissues during postnatal life [125,126,127].

### 4.4. Angiogenesis

Angiogenesis refers to the growth of new capillaries from the preexisting vasculature through sprouting and remodeling of the generated vascular tubes [123]. Physiological angiogenesis is a normal biological response that is happening in several physiological conditions such as wound healing and in the female reproductive cycle. In contrast, pathological angiogenesis is mostly observed during persistent inflammatory conditions, vascular retinopathies (retinopathy of prematurity or diabetes-associated retinopathy), and tumor growth [128]. It also has been proposed that while vasculogenesis is probably responsible for the initial process of retinal vascular development [129], angiogenesis becomes more dominant in the formation of the vasculature in the retina [121].

As described above, elaboration of the retinal vascular network takes place mainly through physiological angiogenesis, and may be explained by the endothelial tip-stalk model. Degradation of the basement membrane, detachment of pericyte cells, and loosening of endothelial cell junctions are important biological steps that occur before the initiation of sprouting angiogenesis in the retina [123]. Cells at the tip of the sprout are more motile and less proliferative, while cells in the stalk are more proliferative and form tubes and branches [130]. Tip cells with their numerous projecting filopodia sense gradients of VEGF and other angiogenic factors (semaphorins, netrins, and ephrins) and guide the direction of the nascent blood vessels to an appropriate final destination [123]. In contrast, stalk cells form the vessel lumen and sustain sprout extension [130].

The tip or stalk state in endothelial cells is extremely transient and reversible [123]. Endothelial tip cells repress the tip phenotype in the neighboring cells via an integrated feedback loop between the VEGF and NOTCH signaling pathways. In tip cells, VEGF/VEGFR-2 signaling promotes DLL4 expression [131]. DLL4-mediated activation of the NOTCH signaling pathway in neighboring endothelial cells suppresses the tip phenotype and drives endothelial cells to the stalk state [123]. However, the promoted expression of JAG1 (NOTCH ligand) in stalk cells antagonizes DLL4 activity in the adjacent tip cells [123]. Recent evidence has also demonstrated the importance of BMP9/10-ALK signaling in suppressing the tip state in endothelial cells [132,133,134,135,136].

### 4.5. Retinal Vasculature in Eye Diseases

Breakdown of the blood vessel networks is the most common cause of loss of sight in industrialized countries. In 2020, more than 43 million people lived with blindness, and the prevalence of vision impairment is predicted to worsen by 2050 [137], when 61 million people are expected to be blind and 474 million people will suffer from moderate to severe vision impairment [137]. Pathological retinal neovascularization has been reported in a broad spectrum of eye disorders including retinopathy of prematurity (ROP), diabetic retinopathy (DR), and age-related macular degeneration (AMD). Abnormal ocular angiogenesis in the retina is associated with the formation of leaky and tuft-like vessels [122]. Leakage of blood from disorganized vessels reduces the amount of oxygen and nutrients that get to the retinal tissue and results in impaired vision and blindness. The generated hypoxia–ischemia condition in the retinal tissue also stimulates the release of angiogenic factors and promotes vascular growth.

#### 4.5.1. Retinopathy of Prematurity (ROP)

Retinopathy of prematurity (ROP) is a major cause of blindness in the pediatric population which is responsible for around 6–18% of cases of blindness in the United States [138,139]. The development of human blood vessels starts during the fourth month of gestation and reaches the retinal periphery just before birth. So, the retinas of infants born prematurely are characterized by incomplete vascularization and a peripheral avascular zone at birth [140]. The pathogenesis of ROP has been described in two opposite phases and both oxygen-regulated and non-oxygen-regulated factors are involved in normal vascular development and retinal neovascularization. The entity of opposite phases in ROP pathogenesis is accentuating the timing of pharmaceutical treatment or any clinical intervention.

Phase I of ROP is characterized by vessel loss occurring from birth to the post-menstrual age (PMA), typically around 30–32 weeks [140]. This phenomenon is thought to be due in part to the relative hyperoxia of the extrauterine environment, compared to the in utero environment [122]. Ultimately, hyperoxia halts the secretion of angiogenic factors that cease the normal growth of the retinal vasculature and result in the loss of some developed vessels [140]. Phase II of ROP is identified by hypoxia-induced vascular proliferation and begins at about 32–34 weeks of PMA [140]. The absence of an adequate retinal vascular system in prematurely born infants results in tissue hypoxia and secretion of hypoxia-driven angiogenic growth factors [122]. In this phase, the formation of the leaky blood vessels at the junction between the non-vascularized retina and vascularized retina can cause retinal detachment and blindness [140].

#### 4.5.2. Diabetic Retinopathy (DR)

Diabetic retinopathy (DR) is the leading cause of blindness in working aged adults and is clinically divided into non-proliferative and proliferative stages. Typically, non-proliferative DR, which can progress to proliferative DR, is diagnosed based on the presence of retinal vasculature abnormalities including microaneurysms, dot and blot hemorrhages, cotton wool spots, edema, and capillary nonperfusion owing to microvascular damage and pericyte loss [122,141]. However, proliferative DR is typically diagnosed based on the neovascularization on the surface of the retina [141]. In this stage, capillary vaso-obliteration and capillary occlusion contribute to the development of retinal ischemia which triggers the secretion of hypoxia-induced growth factors (e.g., VEGF) and promotes pathological angiogenesis in patients with diabetic retinopathies [122,141]. Pathological neovascularization in proliferative diabetic retinopathy results in vitreous hemorrhages, retinal detachment, and finally, vision loss [142,143].

#### 4.5.3. Age-Related Macular Degeneration (AMD)

Age-related macular degeneration (AMD) is the leading cause of irreversible vision loss in aged people over 55 years which is responsible for 6–9% of blindness worldwide [144,145]. Unfortunately, it has also been estimated that almost 288 million people will suffer from AMD in 2040 [144]. AMD is a multifactorial disease that is characterized clinically by the accumulation of drusen and progressive degeneration of photoreceptors and retinal pigmented epithelium (RPE) [146]. Drusen are extracellular deposits of lipids, proteins, and minerals that accumulate under the retina [146]. A vascular etiology has been suggested for the pathogenesis of AMD. In the early stage of disease, a reduction in choriocapillaris density through loss of endothelial cells removes the vascular support of the retinal pigmented epithelium and stimulates the secretion of angiogenic factors [147]. Then, excessive growth of abnormal blood vessels from the choriocapillaris (choroidal neovascularization) penetrates through Bruch’s membrane into the subretinal pigment epithelium and results in vision loss due to the formation of a fibrotic scar and detachment of retinal pigmented epithelium [122,147].

## 5. Implications of SCs on the Retinal Vascular System

The retina is one of the most energy-demanding tissues in the human body, and its structural and functional integrity chiefly depends on a regular oxygen/nutrient supply [148,149]. Dysregulated angiogenesis and sustained imbalances in the supply of nutrients and oxygen stimulate stress responses in vital tissues of the retina such as cellular senescence [150]. Upregulation of classical senescence-associated markers including p53, p16^INK4a^, plasminogen activator inhibitor 1 (Pai1), Cdkn1α (p21^(CIP/WAF1)^), and promyelocytic leukemia protein (PML) provide further evidence for the presence of senescent retinal cells in retinopathies [149]. However, the response of nervous and vascular cells to the hypoxic/oxidative nature of ischemic retinopathy is not homogeneous across all retinal layers. At P14, cellular senescence is confined to avascular regions and then spreads to pathological vascular tufts and retinal microglia [149]. Apoptotic cell death occurs more frequently in cells of the inner nuclear layer (INL); however, cells of the RGC layer (GCL) adopt a senescent phenotype [149]. 

Here, we review the implications of cellular senescence, with a particular emphasis on the retinal vascular system and how SCs might reside in tissues by escaping from immune-mediated clearance.

### 5.1. Detrimental Effect: Vascular Degeneration

Retina blood vessel is made up of three distinct layers of the tunica intima, the tunica media, and the tunica adventitia. The tunica adventitia consists of fibroblast cells, while the tunica media is hosting smooth muscle cells. The tunica intima, the thinnest layer of the vascular wall, is composed of one layer of endothelial cells (ECs) and a basement membrane. ECSs not only act as a physical barrier but also participate in the regulation of vascular homeostasis, vessel tone, inflammatory responses, and neovascularization [151]. A previous study by Bertelli et al. demonstrated that long-term high glucose exposure stimulates premature senescence in human retinal ECs [152]. Senescent ECs displayed limited replicative potential and increased susceptibility to pathological attacks [153]. Losing replicative potential inhibits cell endothelialization and declines the repair function of vessels [153]. The presence and accumulation of senescent vascular cells also results in a leaky and highly permeable endothelium, allowing immune cells and circulating non-immune cells to enter the surrounding tissue and provoke a low-grade inflammatory state [154,155,156]. In both in vivo and in vitro studies of the blood-brain barrier (BBB) model, the accumulation of senescent vascular cells is accompanied by lower barrier integrity [157]. A study by Venkatesh et al. also demonstrated that EC senescence is associated with increased permeability due to the alteration of vascular endothelial (VE)-cadherin and β-catenin expression and localization [158]. Krouwer et al. provided further evidence for the detrimental effect of senescent ECs where the presence of replicative senescence in a non-senescent endothelial monolayer impaired both adherence and tight junctions and compromised the integrity of the endothelial barrier [159]. Altogether, these studies indicate that the senescence phenotype of ECs is associated with lower barrier integrity in retinal blood vessels (Figure 4).

### 5.2. Detrimental Effect: Pathological Angiogenesis

The hypoxic/oxidative nature of the ischemic retina triggers cellular senescence predominantly in the avascular zone to protect retinal cells from a low metabolic supply and hypoxia-associated cell death [149]. The senescence phenotype is not limited just to the avascular zone and there is a wealth of evidence showing the SASP’s engagement in propagating cellular senescence to the surrounding tissue in an autocrine and paracrine manner [54,160,161]. For instance, during the progression of retinopathy, secretion of the neuron-derived Semaphorin 3A (SEMA3A) by senescent retinal neuronal cells contributes to propagating paracrine senescence [149]. A recent study by Crespo-Garcia, Tsuruda, and Dejda et al. also demonstrated a cellular senescence burden at the peak of retinal neovascularization in a mouse model of oxygen-induced retinopathy (OIR) [162]. The high metabolic activity of the stimulated retinal SCs eventually promotes inflammation in the surrounding tissue microenvironment via the secretion of a pool of bioactive molecules (SASPs). These molecules include matrix-degrading proteases, growth factors, inflammatory chemokines, and cytokines. Further research by Oubaha et al. demonstrated that the ischemic retinal cells in retinopathies become prematurely senescent and exacerbate pathological angiogenesis by secreting a series of inflammatory cytokines [149]. Altogether, senescent retinal cells not only contribute to propagating the senescence phenotype in retinopathies, but also promote the destructive growth of new blood vessels from preexisting vessels (Figure 5).

### 5.3. Beneficial Effect: Vascular Remodeling

Vascular development and remodeling occur in the retina by executing two predefined physiological programs: (1) vascular growth (sprouting, proliferation, and branching (see Section 4) and (2) vascular regression. In the fetal developing eyes, the programmed regression of vessels happens during the development of the hyaloid vasculature system. Failure in the regression of the hyaloid vessels causes severe intraocular hemorrhage, massive accumulation of vessels, and leads to persistent hyperplastic primary vitreous disease (PHPV) [163]. Physiological regression of blood vessels is not limited just to the retina but has also been discovered in the ductus arteriosus [164] and female reproductive system (during endometrial maturation) [165]. Vascular remodeling and regression were also detected in pathogenesis studies of several retinal disorders. Proliferative retinopathies (ROP, DR, and AMD) are associated with the pathological formation of a network of new blood vessels that are leaky, fragile, tortuous, and misdirected (see Section 4.5). Recent studies demonstrated the necessity of pruning pathological neovessels to prepare the retina for reparative vascular regeneration [149,166]. The spontaneous reparative vascular regression at the late stage of diseases usually attenuates the destructive costs and implications of the pathological neovessels [167]. This is supported by the fact that almost 90% of all affected ROP infants in the United States recover spontaneously without any therapeutic intervention [166,168,169]. Spontaneous regression of the pathological neovessels was also reported in DR individuals [170,171]. However, the cells and underlying mechanisms mediating vascular remodeling and regression in the retina have not been well defined.

Vascular development and homeostasis is highly dependent upon a healthy endothelium [172]. The endothelium, a monolayer of vascular ECs, releases substances that are key modulators of matrix remodeling, vascular tone, inflammatory responses, and vascular cell proliferation [172]. Endothelial dysfunction is a common finding in all major cardiovascular diseases such as hypertension, atherosclerosis, and diabetes [173]. Recent studies also indicated that the metabolism of ECs regulates vascular growth (angiogenesis and vasculogenesis), and their central metabolism rewires during pathological vessel overgrowth [174]. Yoshikawa et al. (2015) even discovered that the regression rate of hyaloid vessels highly depends upon endothelial VEGFFR2 signaling [163]. It is therefore tempting to speculate that endothelial cells are possibly involved in the mechanisms mediating vascular remodeling and regression in the retina. Computational studies have now confirmed the enrichment of senescent markers in endothelial cells when preretinal neovascularization reaches its maximum level at P17 [149,162,166]. Senescence-associated secretory factors (e.g., TNF-α, IL-6, and MCP-1, and MMP) also play a leading role in matrix remodeling in vascular development and diseases [173].

SCs are resistant to apoptosis and generally are cleared by the immune system instead of necrotic or apoptotic mechanisms. The beneficial effect of the immune-mediated clearance of SCs in the progression of several developmental and physiological processes has been described recently (see Section 3). Impairment in the mechanisms of the SCs’ removal might result in cancer or aging-related disorders [175]. To date, a specific subpopulation of immune cells was shown to be recruited by SCs via SASP factors, this recruitment being essential to the elimination of SCs. For instance, in a model of liver fibrosis, senescent hepatic stellate cells promoted the resolution of fibrosis by attracting natural killer (NK) cells [101]. Antigen-specific CD4+ T cells orchestrate senescence surveillance of pre-malignant hepatocytes and prevent the development of murine hepatocellular carcinomas (HCCs) [176]. Abnormal accumulation of SCs in the postpartum uterus may impair subsequent pregnancy in mice [175]. Tissue-resident macrophages are another component of the immune system that is involved in the clearance of SCs. Egashira et al. (2017) provided evidence for the involvement of F4/80+ macrophages in the clearance of postpartum uterine senescence during the physiological process of uterine remodeling [175].

Similarly, in the retina, the transient presence of SCs is also beneficial, largely by recruiting immune cells for their elimination, but the prolonged presence of SCs can be deleterious (see Section 5.1 and Section 5.2). In ischemic retinopathies, the pathological neovascularization phase is followed by a phase of vascular remodeling and regression. However, the underlying mechanisms of the vascular remodeling and regression in the retina are ill-defined. Long-term exposure to the hyperglycemia, hypoxia, and inflammatory stress induces damage to the neurovascular cells including endothelial cells and pericytes [168]. The leaky, misdirected, and abnormal formed vessels in the neovascularization phase is not capable to fully supply the high oxygen and nutrient demands of the retina, leads to the generation of a tissue microenvironment with profound oxidative, cellular, and inflammatory stress [168]. A growing body of evidence supports the significance of the inflammation and pro-inflammatory mediators in driving proliferative retinopathies and pathological neovascularization [177,178,179,180]. Stressed retinal cells boost local inflammation and recruit immune cells to the ischemic retina [168]. Stressed endothelial cells, for example, enhance leukocyte adhesion and transmigration in microvessels via upregulating adhesion molecules (ICAM-1, VCAM-1, etc.) [181,182]. Recently, it has attracted much attention to explore the possible role of immune cells in the regression of pathological retinal neovessels (tufts). Recruited regulatory T cells to the ischemic retina repaired pathological neovascularization probably by altering activation state of microglia [177]. Inflammation and cellular stress in the retina stimulates activation and infiltration of mononuclear phagocytes (MPs) in the ischemic retina [168]. In this regard, studies in OIR model and in DR confirmed elevated number of MPs in the retina tissue [183,184]. Time dependent studies of the MPs distribution demonstrated that the number of M1-polarized MPs elevates during the phase of pathological neovascularization (starts at P12 and peaks at P17) while M2-polarized MPs population increases during the phase of vascular remodeling and regression (starts at P17 and peaks at P20) [185,186]. Recent studies suggested that MPs attenuate neovascularization and resolves existing pathological neovessels probably via killing (FasL-Fas interaction) and phagocyting stressed endothelial cells at pathological neovascular tufts [168,187,188,189,190]. Findings by Davies et al. also suggest that local secretion of MPs-mediating chemokines, such as MCP-1, promotes the resolution of pathologic neovessels by attracting retinal MPs to the neovascular tufts [191]. An interesting and recent study by Binet et al. led to important discoveries: (1) an enrichment of senescence and SASP markers in pericytes and ECs, (2) the arrival of neutrophils at the sites of pathological neovascular tufts when vascular regression begins (at P17), and (3) the detection of neutrophil extracellular traps (NETs) adjacent to the pathological neovessels [166]. Indeed, senescence-associated secretory phenotype of senescent ECs (e.g., IL-1β and CXCL1) attracts neutrophils to the pathological neovascular tufts and prompts the release of NETs [166]. Subsequently, NETs induce apoptosis in senescent ECs and promote regression of the pathological neovascular tufts in the ischemic retina [166] (Figure 6). Taken altogether, these findings demonstrated that inflammation and transient present of SCs at the late stage of proliferative retinopathies promotes regression of pathological neovessels and prepare ischemic retina for reparative vascular regeneration.

Although NETosis is beneficial for clearing diseased senescent vasculature, excessive NET formation may contribute to the development of conditions such as diabetes and its complications, like diabetic retinopathies (DRs) [192]. Hyperglycemia, both in vivo and in vitro, increases NET formation [193,194]. In addition, a significant surge in circulating NETosis markers (MPO, NE) has been documented in the serum of individuals with proliferative diabetic retinopathy (PDR) [195]. The specific mechanism by which high glucose induces NET formation in DR patients and how NET formation contributes to the pathogenesis of DR has not been fully elucidated. However, a recently published study by Wang et al. has proposed that long-term hyperglycemia induces the formation of NETs through an NADPH oxidase-dependent pathway in diabetic patients [192].

## 6. Therapeutic Targeting of Senescent Cells

SCs are resistant to apoptosis and accumulate in aged tissue over time. The accumulation of SCs in many age-related disorders [196] and their negative implications in the development of many diseases have led scientists to search for strategies that prevent or disrupt cellular senescence. Documenting the anti-aging properties of the calorie restriction in several species has opened the door for the development of agents and strategies that might extend a healthy lifespan and delay the onset of age-related diseases [197,198]. Since then, many natural products have been isolated from plants or fungal sources and proposed for eliminating SCs [61,199]. At present, the two most utilized therapeutic approaches in targeting SCs are the following: (1) senolytics, which selectively eliminate (kill) SCs, and (2) senomorphics, which modulate the cell-extrinsic effects of the SASP secreted by SCs.

### 6.1. Senolytics

Upregulation of several antiapoptotic pathways, including members of the Bcl-2 family (including Bcl-2, Bcl-w, and Bcl-xl), has made SCs resistant to apoptosis and to their own proapoptotic, tissue-destructive SASP [182]. In contrast, non-SCs are unprotected against the harmful effects of SASP, and therefore, targeting antiapoptotic pathways to induce apoptosis in SCs is the strategy many senolytic compounds use to eliminate SCs selectively.

Dasatinib (D) and quercetin (Q) were the first identified senolytic compounds used for tackling SCs in human and animal studies. Quercetin (a member of the flavonoid family) blocks serpins and some kinases including phosphoinositide 3-kinase (PI3K), while dasatinib is a pan-tyrosine kinase inhibitor that also blocks ephrin-dependent receptor signaling [177,178,179,180]. The senolytic effect of dasatinib has been demonstrated in the radiation-induced senescent human pre-adipocytes, while quercetin was more effective against radiation-induced senescent HUVECs and mouse BM-MSCs (bone marrow-derived mesenchymal stem cells) [180,181,182,183]. The combined regimen of dasatinib (D) and quercetin (Q) effectively removed the senescent mouse embryonic fibroblasts (MEFs) and reduced senescent cell burden in chronologically aged, radiation-exposed, and progeroid ${Ercc1}^{\left(-/∆\right)}$ mice [183]. Further research by authors also indicated improved cardiac function and carotid vascular reactivity in aged mice [183]. The first clinical trial of this combination (Q and D) in patients with idiopathic pulmonary fibrosis (IPF) revealed improved physical function; however, pulmonary function, clinical chemistries, frailty index, and reported health remained unchanged [184]. Fisetin is another member of the flavonoid family that induces apoptosis in radiation-induced senescent HUVECs [185] and oxidative stress-induced senescent primary murine embryonic fibroblasts [186]. Fisetin administration in aged wild-type mice was associated with improved tissue homeostasis, suppression of age-related pathologies, and extended median and maximum lifespan [186].

Navitoclax (ABT-263), ABT-737, A1331852, and A1155463 are other senolytic agents belonging to the Bcl-2 family protein inhibitor [180]. Navitoclax (ABT-263), for example, induces apoptosis in SCs via binding to the BH3 domain of Bcl-2, Bcl-xl, and Bcl-w proteins (negative regulators of apoptosis) [180]. Like dasatinib (D) and quercetin (Q), navitoclax is not a cell type-independent senolytic agent [188]. Zhu et al. documented the senolytic effects of navitoclax in HUVECs, IMR90 cells, and MEFs, but not in senescent human primary preadipocytes [187]. Navitoclax (ABT-263) administration to sublethally irradiated mice depleted the SC burden in bone marrow hematopoietic stem cells and muscle stem cells (MuSCs) [188]. Navitoclax-mediated clearance of SCs in ${Ldlr}^{\left(-/-\right)}$ mice with a high-fat diet removed foam cell macrophages and inhibited atherogenesis [189]. Transient thrombocytopenia, neutropenia, and bleeding are the common side effects of navitoclax treatment in patients [190].

### 6.2. Senomorphics

Targeting the SASP without removing SCs is the second therapeutic strategy that uses senescence-related phenotypes or diseases. Senomorphics attenuate the SASP implications through suppressing pathways involved in the induction and maintenance of the SASP (described in Section 2).

Metformin, apigenin, kaempferol, and BAY 11-7082 are several examples of the senomorphics that inhibit SASP via modulating the NF-κB signaling pathway. The antidiabetic drug metformin has been shown to inhibit the activation of the NF-κB, a master regulator of the inflammatory responses, in bovine retinal capillary endothelial cells and retinal endothelial cells from diabetic rats (SIRT1/LKB1/AMPK/ROS pathway) [200]. Further studies by Arunachalam et al. (2013) also demonstrated that metformin protects mouse microvascular endothelial cells (MMECs) from hyperglycemia-induced premature senescence via attenuating the reduction in SIRT1 expression [201].

The pharmaceutical effects of metformin occur through multiple pathways which make it difficult to clearly define the exact mechanism for the antiaging effects of metformin [202]. However, it is suggested that metformin mainly reduces the expression of inflammatory cytokines through the inhibition of the NF-κB signaling pathway [61]. Disturbed NF-κB translocation into the nucleus by metformin prevents IKB and IKKα/β phosphorylation and finally inhibits SASP production [203]. Metformin-mediated activation of the AMPK (AMP-activated protein kinase) has also been found to reduce protein synthesis in cells via inhibiting the mTORC1 signaling pathway [204]. An in vitro study has also shown that a chronic low-dose metformin treatment extends the lifespan in human diploid fibroblasts and human mesenchymal stem cells (MSCs) via Nrf2-mediated transcriptional upregulation of ER-localized GPx7 [205]. It also inhibits ROS generation, γ-H2AX foci, and ATM [206,207]. Metformin treatment inhibits the interaction of the decay promoting RBP AUF1 with DICER1 mRNA, thus increasing DICER1 expression in mice as well as in human cells [208]. Metformin-mediated increases in DICER1 expression finally decreased cellular senescence in a DICER1-dependent manner [208]. Metformin administration also decreased the protein levels of p16^INK4a^, p21^(CIP/WAF1)^, and several critical SASP factors including IL6, IL8, CXCL1, and CXCL2 [208]. To sum up, many recent studies have discovered possible targets of metformin action in the context of aging and aging-related disorders; however, a complete description of metformin’s mechanism of action and how these molecular targets are interconnected or work independently is still required.

Rapamycin is another very popular senomorphic agent that suppresses SASP via multiple pathways, mainly mTOR signaling pathway. The mTOR signaling pathway modulates the secretory activity of SCs and its inhibition reduces the expression of SASP components (IL6, Vcam1, IL12b, IL7, and Icam1) in an Nrf2-independent manner and senescence hallmarks (p16^INK4a^ and p21^(CIP/WAF1)^) in an Nrf2-dependent manner [209]. Rapamycin treatment alleviated senescence burden and increased the lifespan of middle-aged mice [210]. The STAT3 signaling pathway is another SASP regulatory pathway that is targeted by rapamycin administration [61]. The expression of many SASP factors is regulated by the NF-κB signaling pathway (described in Section 2). There is also a positive feedback loop between NF-κB transcription factor and the membrane-bound cytokine IL1A that modulates the transcription of several inflammatory cytokines [79]. As expected, rapamycin treatment inhibited NF-κB transcriptional activity via decreasing IL1A production and ultimately reduced the expression of various SASP factors [79]. However, rapamycin-mediated mTOR inhibition is associated with serious side effects including insulin resistance, glomerular dysfunction, dyslipidemia, hematologic side effects, mucositis, pneumonitis, lymphedema, angioedema, and osteonecrosis [211,212].

## 7. Discussion

As we get older, more cells in healthy tissues become senescent. SCs are inactive in terms of reproduction, but extremely active in terms of metabolism and potentially inflame the milieu by producing thousands of bioactive molecules. Growth and development are not possible without the presence of SCs due to the critical role of SCs in a variety of biological processes such as embryogenesis, limb generation, wound healing, host immunity, and tumor suppression. However, due to the proinflammatory entity of senescent cells, their chronic accumulation is associated with a gradual decline in tissue function and age-related disorders. In this review, an overview of the underlying mechanisms behind SC formation, as well as the different signaling pathways that modulate components of the SASP profile are described.

While both arteries and veins are part of the circulatory system and share some similarities in structure and function, they also have distinct characteristics and roles. Therefore, the implications of SCs on arteries and veins can differ based on several factors, including their location, the type of senescent cells involved, and underlying health conditions. For instance, atherosclerosis is primarily associated with arteries, and the presence of senescent cells in arterial walls can exacerbate this condition, leading to narrowed arteries and reduced blood flow [213]. Similarly, in the venous system, cellular senescence could play a role in the development of chronic venous diseases such as varicose veins [214,215]. However, the implications of SCs on different components of the retinal vascular system (capillaries, arteries, and veins) remain unclear and require further investigation.

Diseased blood vessels are a common feature in many eye disorders including retinopathy of prematurity, diabetic retinopathy, and age-related macular degeneration. Mounting recent evidence has discovered the accumulation of senescent neurons and blood vessels in the retina. However, the underlying mechanisms of senescent cell contribution in retinal vasculopathies are not well defined yet. Here, we reviewed dichotomous implications of SCs at the onset and severity of proliferative retinopathies with a specific focus on the retinal vascular system. In a retinal blood vessel, the senescence phenotype in endothelial cells is associated with lower barrier integrity and increased permeability probably due to the impairment of both adherence and tight junctions. In retinopathies, the hypoxic/oxidative stress induces cellular senescence in retinal neuronal cells that reside predominantly in the avascular zone. The inflammatory secretome of the cell cycle arrested cell boosts and propagates the senescence phenotype to the surrounding tissues in a paracrine and autocrine manner. Dysregulated angiogenesis is another feature of proliferative retinopathies in which SCs play a role. The presence of angiogenic factors, as a part of the SASP secretome, attracts tip cells of retinal blood vessels to the ischemic area and leads to excessive uncontrolled vascularization in the retina. The newly formed blood vessels are leaky, tortuous, and misdirected and do not properly supply the high energy-demanding tissues of the retina and stimulate the senescence phenotype in surrounding retinal cells.

In the retina, it is vital to bear in mind that all implications of SCs are not detrimental. Immune-mediated clearance of senescent endothelial cells at the late stage of proliferative retinopathies promotes regression of the pathological neovascular tufts and prepares the retina for reparative vascular regression. Recruited mechanisms by retinal immune cells for eliminating stressed endothelial cells are comprehensively described in this review. Finally, senolytics and senomorphics are discussed as two main available therapeutic strategies for eliminating retinal SCs in proliferative retinopathies.

Several recently published studies established a firm connection between cellular senescence and proliferative retinopathies [149,166]. However, many aspects of the interplay between retinal SCs and retinal vasculopathies still remain to be illustrated and need further research. According to the nature of stimuli/stress, different kinds of SCs might be present in any given tissue. The SASP components of each senescent cell might be different due to using different pro-survival pathways. Indeed, the high heterogeneity of SCs at the single cell level makes our understanding of the SCs’ implications in tissue and therapeutic targeting more and more sophisticated. Many components of the SASP profile are inflammatory factors and their elevated expression is not sufficient and definitive for detecting SCs. Indeed, the lack of distinctive and cell-specific biomarkers for the straightforward detection of senescent cell subpopulations is another fundamental challenge in this field of research. However, recent advances in single-cell transcriptomic approaches such as spatial transcriptomics have provided a potential and promising tool for finding novel biomarkers of senescent subpopulations and an opportunity for addressing the similarity and complexity of SCs. In this review, the dual implications of SCs in the retina emphasize the notion of discerning physiological SCs from pathological SCs to promote the efficiency of our therapeutic interventions via suppressing harmful functions while permitting beneficial functions of SCs. Unfortunately, current senotherapeutic strategies do not discriminate physiological SCs from pathological ones and are associated with multiple side effects. Therefore, expanding our knowledge about senescence mechanisms and SASP modulating pathways will candidate novel cell-specific targets for senotherapeutic treatments without eliminating physiological/developmental SCs.

## 8. Conclusions

SCs are characterized by irreversible cell cycle arrest and heterogeneity of the senescence phenotype. The complexity of the regulatory mechanisms of SASP are outlined. Recently, many published studies have disclosed a robust association between cellular senescence and age-related disorders as well as physiological conditions. Indeed, it appears necessary to examine the overall implications of SCs in individual tissues by considering two distinct perspectives: beneficial and detrimental roles. Likewise, for the enhancement of therapies targeting proliferative retinopathies, it could be paramount to gain insight into SCs in the retina, considering both their beneficial and detrimental effects. Here, a literature review of the previously published studies demonstrated how the permanent residence of SCs in the retina may promote comorbidities in proliferative retinopathies through vascular degeneration or pathological angiogenesis. In contrast, the transient presence of SCs at the late stage of PR enhances the regression of pathological neovessels to prepare a stressed retina for reparative vascular regeneration. Nonetheless, numerous questions regarding the potential role of SCs in the progression and resolution of proliferative retinopathies remain unanswered. In consequence, further investigation into the mechanisms underpinning the pathways and processes that mediate the progression of retinopathy is needed to provide a better understanding of how to target SCs more efficiently in the context of proliferative retinopathy, while preserving the beneficial functions of transient senescent cells in the retina.

## Figures and Tables

**Figure 1 cells-12-02341-f001:**
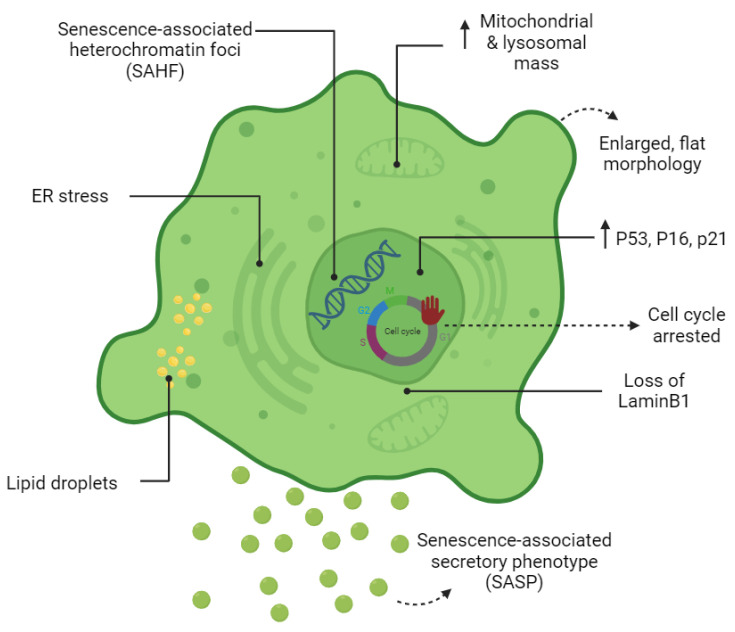
Cardinal features of senescent cells. The morphology, metabolism, and biomarkers of senescent cells are illustrated.

**Figure 2 cells-12-02341-f002:**
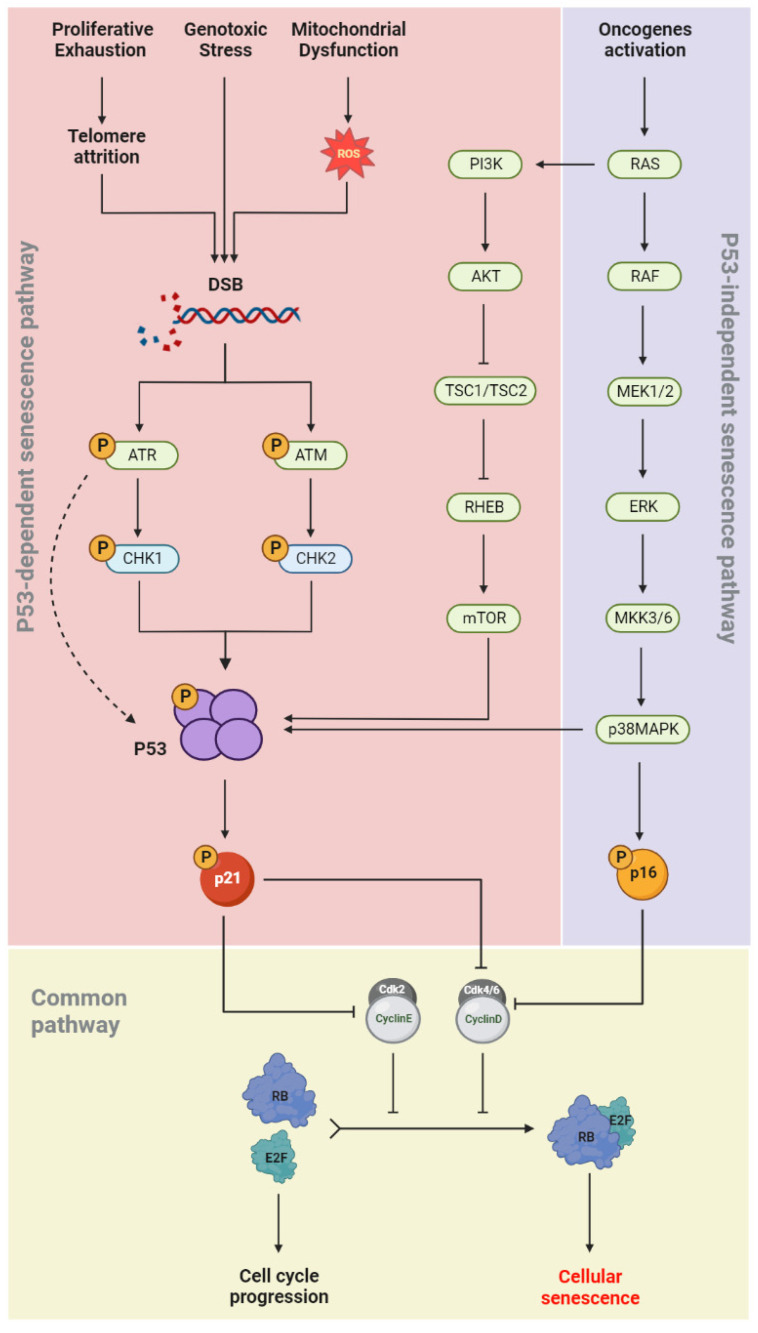
Senescence molecular pathways. Cell cycle arrest occurs in senescent cells via two independent pathways: p53-dependet senescence pathway and p53-independent senescence pathway. Senescence inducers and the main effective downstream molecules are illustrated.

**Figure 3 cells-12-02341-f003:**
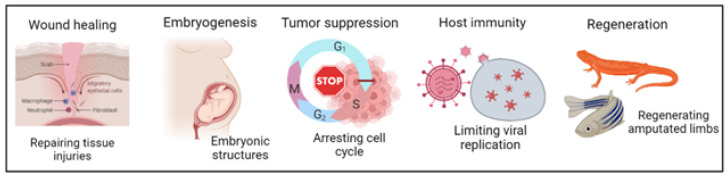
Beneficial effects of SCs. Just the three beneficial effects (embryogenesis, regeneration, and wound healing) that are more related to the context of this review were discussed in the main text.

**Figure 4 cells-12-02341-f004:**
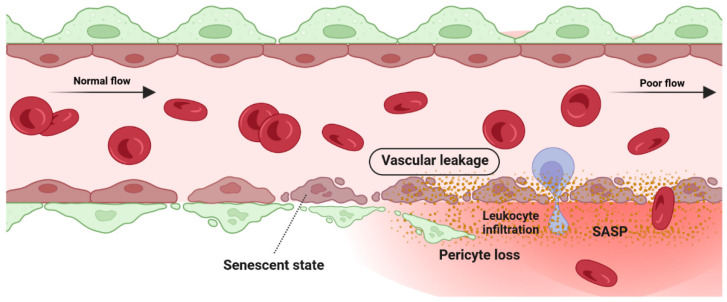
Detrimental effect: vascular degeneration. Senescence phenotype in pericyte and retinal endothelial cells is associated with pericyte detachment and vascular leakage.

**Figure 5 cells-12-02341-f005:**
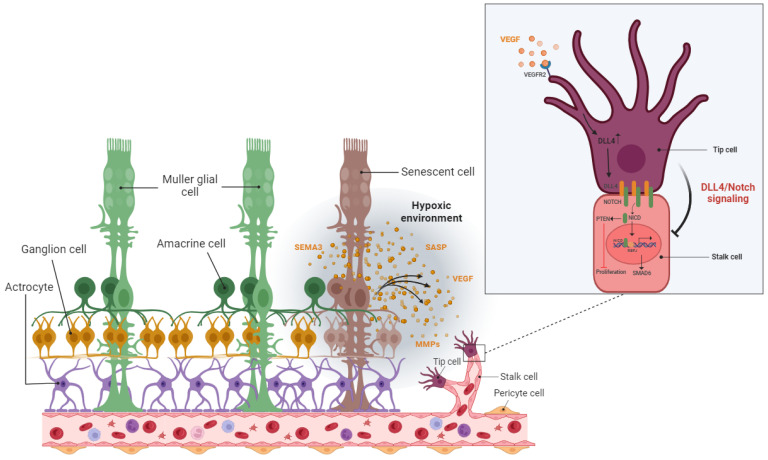
Detrimental effect: pathological angiogenesis. In the stressed retina, senescence neurons in the avascular area attract tip cells of the retinal blood vessel by releasing various proinflammatory and angiogenic factors and lead to dysregulated angiogenesis as a common feature of proliferative retinopathies.

**Figure 6 cells-12-02341-f006:**
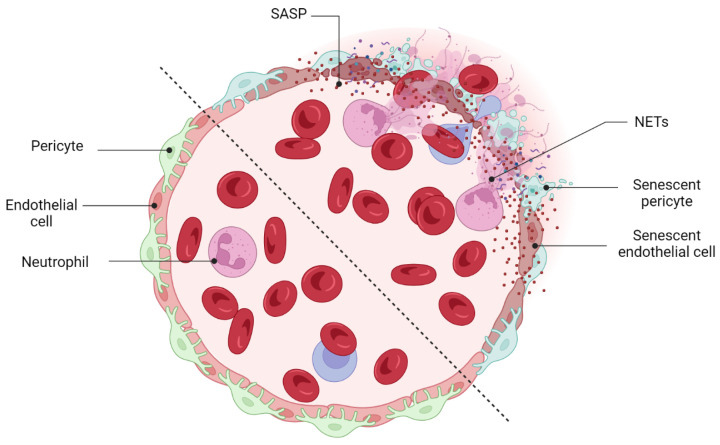
Beneficial effects of cellular senescence: vascular remodeling. Immune-mediated clearance of retinal SCs promotes the regression of pathological neovessels. Neutrophils recruit NETs (neutrophil extracellular traps) and prepare the retina for a reparative mechanism of regeneration by eliminating tuft-like neovessels.

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
