# Peer review of "Senescent Cells: Dual Implications on the Retinal Vascular System"

_cells, 2023, doi:10.3390/cells12192341_

Round 1

Reviewer 1 Report

General comments

The authors present a review of senescence in cells (SC) (first section) and then focus on this phenomenon in the context of retinal vascular disease (second section). The review is well written overall and details the different components of cell function in SC. This is an interesting report and the authors have highlighted a number of conundra. They emphasise that senescence is not a forerunner of cell death be it apoptosis or necrosis / pyroptosis and in fact may even be integral to any physiological process, including embryogenesis, tissue remodelling, and aging. They have, however, not clearly argued that SC is a distinct process with a specific outcome for the cell, separate from the accepted life cycle of every cell, which is of course very different for each cell and is dependent on the intrinsic replication rate. For instance, where do "terminally differentiated" cells fit in their cellular life scheme? What are quiescent but non-senescent cells? Is senescence triggered by external signals or is it dependent on, for instance, clock genes?

Also, the process of senescence appears to be associated with a senescence associated secretome which includes several proteins involved in inflammation, such as IL6 and IL1b. Could this explain "para-inflammation" which is associated with aging and is not necessarily detrimental to health?

With regard to the role of SC in retinal vascular disease, the authors present a valid case. There are a number of aspects which require clarification: for instance is SC different for larger vs smaller (capillary) vessels, arterioles vs venules; do senescent Muller glia cells initiate damage in retinal vessels in diabetic retinopathy; is there evidence for neutrophil traps in retinal vascular disease especially diabetic retinopathy?

Specific comments

·      is failure of cytokinesis with multinucleation a component of SC?

·      are SC multinucleated cells more or less metabolically active than their mononuclear counterpart eg in retinal pigment epithelium?

·      Is failure of autophagy a part of SC?

·      when cells shift irreversibly to aerobic glycolysis on activation, is this the beginning of SC or cell death? 

·      are senescent fibroblasts in control of successful wound healing?

·      DR is characterised by microaneurysms in the retinal vessels: do the authors consider this to be failed angiogenesis or regressed tip cells due to SC?

·      Section 4.2, line 379: please specify which species this sentence refers to.

·      There are a number of spelling, grammatical and syntactical errors on the following lines: 186, i87, 203, 246, 250, 263, 296, 301, 307, 310, 346, 354, 456, 512, 518, 643, 646

·      references on line 803 are not correctly formatted 

·      reference 110 is incomplete

·      reference 127 is incomplete / incorrectly cited.

Reviewer 2 Report

Comments

In this review, Habibi-Kavashkohie et al. discuss cellular senescence, which leads to structural, metabolic, and functional changes in tissues, accompanied by inflammatory gene expression. Senescent cells accumulate in the retina due to stress and aging, potentially causing chronic diseases and tissue dysfunction through the senescence-associated secretory phenotype (SASP). The review covers the impact of senescence on retina vascular development and retinopathies, while exploring potential senotherapeutic strategies targeting the SASP.

Their reviewing is of great importance and would be of interest to Cells readers. I'd like to propose several minor points that might need to be done to further improve the manuscript.

Minor points:

1) Labels "P51" and "P16" in figures 1 should be "p53" and "p16"?

Also please check "P16" in Line 114, 291, and 494.

2) In line 138, "MTOR inhibitor" means "mTOR inhibitor"?

3) Please note that the correct way to refer to this protein is "p21(CIP1/WAF1)". The various forms you mentioned (e.g., p21, p21(CIP1/WAF), p21(CIP1/WAF), p21CIP/WAF, p21CIP1...etc) are alternate abbreviations or notations for the same protein, and just "p21" or "p21(CIP1/WAF1)" is the generally accepted and standardized nomenclature.

4) In line 490, "struc-tural"

5) All labels including "Muller grial cell" and "Amacrine cell" in figures 5 are a little small. I strongly recommend increasing their font size.

6) In discussion, description in line 767-775 is too introductive and not suitable for discussion. This paragraph should be removed or discussed somewhere.

7) In line 777-778, there are multiple repetitions of abbreviation explanations (e.g., retinopathy of prematurity (ROP)).

8) In lie 803, please check citation format.
